# Options for Topical Treatment of Oxidative Eye Diseases with a Special Focus on Retinopathies

**DOI:** 10.3390/medicina60030354

**Published:** 2024-02-21

**Authors:** Cristina Russo, Dario Rusciano, Rosa Santangelo, Lucia Malaguarnera

**Affiliations:** 1Department of Biomedical and Biotechnological Sciences, University of Catania, Via Santa Sofia, 97, 95124 Catania, Italy; lucmal@unict.it; 2Fidia Ophthalmics, 95124 Catania, Italy; drusciano55@gmail.com; 3Department of Medicine and Health Sciences, University of Catania, Via Santa Sofia, 97, 95124 Catania, Italy; rosa.santangelo@unict.it

**Keywords:** retinopathies, oxidative stress, eye drops, lipid nanoparticles, alkylation

## Abstract

Antioxidants, usually administered orally through the systemic route, are known to counteract the harmful effects of oxidative stress on retinal cells. The formulation of these antioxidants as eye drops might offer a new option in the treatment of oxidative retinopathies. In this review, we will focus on the use of some of the most potent antioxidants in treating retinal neuropathies. Melatonin, known for its neuroprotective qualities, may mitigate oxidative damage in the retina. N-acetyl-cysteine (NAC), a precursor to glutathione, enhances the endogenous antioxidant defense system, potentially reducing retinal oxidative stress. Idebenone, a synthetic analogue of coenzyme Q10, and edaravone, a free radical scavenger, contribute to cellular protection against oxidative injury. Epigallocatechin-3-gallate (EGCG), a polyphenol found in green tea, possesses anti-inflammatory and antioxidant effects that could be beneficial in cases of retinopathy. Formulating these antioxidants as eye drops presents a localized and targeted delivery method, ensuring effective concentrations reach the retina. This approach might minimize systemic side effects and enhance therapeutic efficacy. In this paper, we also introduce a relatively new strategy: the alkylation of two antioxidants, namely, edaravone and EGCG, to improve their insertion into the lipid bilayer of liposomes or even directly into cellular membranes, facilitating their crossing of epithelial barriers and targeting the posterior segment of the eye. The synergistic action of these antioxidants may offer a multifaceted defense against oxidative damage, holding potential for the treatment and management of oxidative retinopathies. Further research and clinical trials will be necessary to validate the safety and efficacy of these formulations, but the prospect of antioxidant-based eye drops represents a promising avenue for future ocular therapies.

## 1. Introduction

The eye’s structure interfaces with the external milieu, rendering it susceptible to environmental and metabolic oxidative stress due to the continuous generation of free radicals. This exposure may lead to structural and functional alterations, giving rise to conditions such as glaucoma, macular degeneration, diabetic retinopathy, dry eye disease, and retinal dystrophies [1,2]. Mitochondria serve as the primary generators of intracellular oxidants, alongside other sources such as nicotinamide adenine dinucleotide phosphate (NADPH) oxidases. However, the capability for intracellular reactive oxygen species (ROS) production extends beyond mitochondria; other cellular organelles, including peroxisomes and the endoplasmic reticulum, also contribute to this process. Additionally, enzymes such as xanthine oxidase, nitric oxide synthase (NOX), cyclooxygenases, and lipoxygenases play a role in intracellular ROS production. ROS and reactive nitrogen species (RNS) primarily affect cellular structures such as proteins, lipids, and deoxyribonucleic acid (DNA), resulting in damage at various cellular levels. Despite this potentially harmful activity, basal levels of ROS are crucial for certain cellular functions, including signal transduction pathways, gene expression, defense against bacterial invasion, and cellular growth or death. Endogen protective agents against ROS are released in the body, such as glutathione peroxidase, superoxide dismutase, or non-enzymatic compounds such as vitamin D, vitamin E, glutathione (GSH), and nicotinamide.

Dysregulation of antioxidant protection is associated with aging and may contribute to the development of most pathological conditions. Within the field of ophthalmology, various studies have reported that, under specific conditions, free radicals may induce chronic inflammatory reactions, altering the physiology of the eye and the visual system [3]. Specifically, ROS can regulate the activity of the transcription factor NF-kB, leading to the upregulation of pro-inflammatory cytokines such as IL-1, IL-6, and TNF-α. Furthermore, the formation of ROS resulting from impaired mitochondria can activate the NOD-like receptor protein 3 (NLRP3) inflammasome, subsequently triggering the release of inflammatory cytokines such as IL-1β and IL-18 [4].

Certain phytochemical compounds, known for their anti-inflammatory and antioxidant properties, such as resveratrol, curcumin, and vitamin D, can be utilized in the prevention or treatment of diseases associated with oxidative stress [5,6,7].

Numerous clinical studies have explored the potential of oral treatment with food supplements to alleviate oxidative eye diseases, indicating the role of specific nutrients in preventing or slowing their progression. The Age-Related Eye Disease Study (AREDS) and AREDS-2 investigated the effects of nutritional supplements on age-related macular degeneration (AMD), a major cause of vision loss in older adults. AREDS demonstrated that a combination of antioxidant vitamins (C and E) and zinc reduced AMD progression and vision loss by about 25% [8]. AREDS-2 found that lutein and zeaxanthin were beneficial, especially for individuals in advanced stages and/or with low dietary intake [9]. Separate studies emphasized the protective roles of lutein, zeaxanthin, and omega-3 fatty acids in retinal health. In aging retinal tissue, a decline in natural antioxidant capacity, specifically a reduction in macular xanthophylls like lutein, zeaxanthin, and mesozeaxanthin, plays a significant role in AMD progression. Considering this, incorporating carotenoid phytochemicals as supplementary therapy in AMD treatment seems reasonable, offering neuroprotection and positive impacts at various AMD stages, including advanced AMD [10]. Omega-3 fatty acids, primarily found in fatty fish, may also protect retinal cells from damage. Associations of xanthophylls and omega-3 have shown benefits in preventing AMD progression and visual function deterioration [11]. Most recently, a newly discovered class of naturally occurring lipid mediators derived from omega-3 very-long-chain polyunsaturated fatty acids (VLC-PUFAs) has been described. Elovanoids (ELVs) play a crucial role in maintaining cellular homeostasis and protecting cells from oxidative stress and damage. More specifically, the mechanism by which ELV-N34 protects neuronal cells from oxidative damage has been found to occur through the inhibition of the thioredoxin enzyme TXNRD1 [12].

Vitamins C and E, with strong antioxidant potential, play a crucial role in protecting cells from oxidative damage. Although individual studies on vitamin C and E supplementation for AMD have yielded mixed results, other studies suggest that a combination of these antioxidants may be beneficial [13]. Several other studies have investigated antioxidant food supplements’ potential to improve vision and slow the progression of diabetic retinopathy in people with diabetes or in patients with glaucoma [14]. The finding that oxidative stress biomarkers may improve after treatment with antioxidants has furthered interest in the clinical application of such treatments.

More recently, attention has shifted toward the specific mechanisms exerted by some specific anti-inflammatory and antioxidant compounds such as melatonin, N-acetyl-cysteine (NAC), edaravone, idebenone, and epigallocatechin-3-gallate (EGCG) (see Graphical abstract).

This review aims to provide a comprehensive overview of the most pertinent and recent evidence regarding the utilization of these compounds, specifically focusing on their potential topical application in treating retinopathies and ophthalmic pathologies resulting from oxidative stress. Additionally, we aim to highlight novel preclinical experimental strategies that can be used to safeguard and enhance the efficacy of these treatments. Given the experimental nature of this approach, which primarily involves in vitro and in vivo model systems, it is evident that the reported results are preliminary. Animal models are known to produce results influenced by their unique eye structure and physiology, which can vary between species and differ significantly from humans [15,16]. Therefore, these results must be confirmed through appropriate clinical trials in humans, as is required for all human therapeutic treatments.

## 2. Oxidative Stress Implications for Retinal Diseases

Oxidative stress plays a significant role in the pathogenesis of various retinal diseases, including diabetic retinopathy (DR), age-related macular degeneration (AMD), glaucoma, and retinopathy of prematurity (ROP). Disruption of mitochondria in photoreceptors and nerves may contribute to the development of such retinal diseases and optic neuropathies. The retina, a crucial structure that generates the optic nerve, receives focused light from the lens, converts it into neural signals, and, finally, reaches the brain occipital cortex for visual recognition. The retina contains two main structures: the neuroretina and the retinal pigmented epithelium. At the center of the retina lies the macula, which is responsible for central vision [17]. The high metabolic activity in the macula exposes it to elevated levels of ROS, generated by the process of vision [18]. The constant exposure of the retina to environmental light, which also contains high-energy UV rays, further contributes to ROS formation [19]. Cumulative and permanent damage caused by electromagnetic radiation impinging on the eye can affect not only the retina but also other ocular tissues such as the cornea, lens, and iris. Such chronic oxidative damage may contribute to the development of various eye pathologies. Vingolo et al. underscored the role of oxidative stress in the progression of ROP due to photoreceptor cell degeneration and dysfunction of the retinal pigmented epithelium [20]. Oxidative stress also induces inflammation of retinal pigmented epithelial (RPE) cells, autophagic cell death, and apoptosis, which are associated with AMD and DR [21,22]. In AMD, oxidative damage to the retinal pigment epithelium (RPE) and photoreceptor cells is a key factor in disease progression. Several studies have underscored the impact of oxidative stress on AMD development and the intricate interplay between genetic susceptibility and environmental factors. Oxidative stress has been shown to be a major contributor to photochemical retinal injury, with antioxidant vitamins A, C, and E showing protective effects against such injury. The formation of lipofuscin, thought to arise from oxidatively damaged photoreceptor outer segments, is associated with retinal damage. While the relationship between dietary and serum levels of antioxidant vitamins and age-related macular disease remains somewhat unclear, the protective role of high plasma concentrations of alpha-tocopherol has been convincingly demonstrated. Macular pigment, which absorbs blue light and may quench reactive oxygen intermediates, is also believed to mitigate retinal oxidative damage [23]. Similarly, in DR, chronic hyperglycemia induces oxidative stress, leading to inflammation, vascular dysfunction, and neuronal damage in the retina. Brownlee’s work has extensively explored the molecular mechanisms linking hyperglycemia, oxidative stress, and diabetic complications, emphasizing the role of advanced glycation end-products (AGEs) and the polyol pathway [24]. Understanding these molecular pathways is essential for the development of targeted therapies aimed at mitigating oxidative stress and preventing the progression of AMD and DR. Of course, oxidative stress is not the sole culprit in eye pathologies. Therefore, antioxidants are, at best, expected to attenuate or delay the progression of such pathologies but do not constitute a cure.

However, the death of photoreceptor cells leads to a progressive decrease in their number, thus resulting in a reduced oxygen demand in the inner retina (since the retinal circulation is not regulated) and toxic hyperoxia in the external retina [25]. In primary open-angle glaucoma (the prevalent form of adulthood glaucoma), the increased intraocular pressure (IOP) and/or microvascular dysfunctions increase oxidative stress and activate inflammatory pathways, finally triggering retinal ganglion cell (RGC) apoptosis through the Bax/caspase-9 pathway [26]. Elevated IOP can activate microglia to release inflammatory cytokines, including interleukins IL1β, IL-6, and TNF-α, leading to nitric oxide (NO) and ROS production [27]. In the central nervous system, activated microglia can damage RGCs [28]. Moreover, ROS may modulate the immune response [29] and alter RGC pathways, adversely affecting their health and promoting neuroinflammation.

Oxidative stress also plays a significant role in the main hereditary retinal dystrophies, such as retinitis pigmentosa (RP), Leber hereditary optic neuropathy (LHON), and Stargardt disease (SD).

***Retinitis pigmentosa*** encompasses a cluster of related retinal disorders characterized by a gradual and progressive loss of vision [30]. In individuals with RP, vision loss unfolds as the light-sensing cells within the retina undergo a gradual deterioration. The symptoms of RP primarily revolve around vision loss. Initially, there is a diminished ability to see in low-light conditions, primarily observed during childhood. Challenges with night vision can impede navigation in low-light settings. As the condition advances, blind spots emerge in the peripheral vision, eventually converging to create tunnel vision. Over the course of years or decades, central vision, crucial for tasks such as reading, driving, and recognizing faces, also succumbs to the disease. In adulthood, many individuals with RP ultimately reach a state of legal blindness. Oxidative stress is considered a significant contributor to the pathogenesis and progression of RP [20,31]. Several mechanisms link oxidative stress to the degeneration of photoreceptor cells in RP. Photoreceptor cells have high metabolic activity, and this leads to an increased production of ROS as byproducts. The excessive generation of ROS can overwhelm the cellular antioxidant defense mechanisms, leading to oxidative stress. Mutations in genes associated with RP can disrupt the balance between oxidants and antioxidants in the retina. For example, mutations affecting the function of antioxidant enzymes or proteins involved in the cellular defense against oxidative stress can compromise the retina’s ability to neutralize ROS. Some forms of RP are associated with mitochondrial dysfunction, leading to an increased production of ROS within mitochondria. Mitochondrial dysfunction and the resulting oxidative stress contribute to the degeneration of photoreceptor cells. Oxidative stress can trigger inflammatory responses and apoptotic cell death in the retina [32]. In RP, the death of photoreceptor cells is often mediated by apoptosis, and oxidative stress is implicated in initiating and amplifying these apoptotic pathways [33]. Oxidative stress can lead to the peroxidation of lipids in cell membranes. In RP, this can affect the integrity of photoreceptor cell membranes, leading to structural damage and cell death. Understanding the interplay between genetic factors and oxidative stress in RP is crucial for developing therapeutic strategies. Some research efforts focus on antioxidants and neuroprotective agents to mitigate oxidative damage and slow down the progression of RP [34]. Additionally, gene therapies aimed at correcting specific genetic mutations associated with RP are being explored as potential treatments [35]. Overall, addressing oxidative stress is a promising avenue for developing interventions to preserve vision in individuals with RP [36].

***Leber hereditary optic neuropathy*** (LHON) is an inherited vision loss disorder with a typical onset during the teenage years or early twenties, with occasional cases emerging in childhood or later adulthood. Notably, males are more frequently affected than females, and the reasons for this gender discrepancy are unknown [37]. LHON initially presents with blurred and cloudy vision, affecting one or both eyes. If one eye is affected initially, the other typically succumbs within weeks or months. Over time, there is a progressive decline in visual acuity and color perception in both eyes, predominantly impacting central vision, which is crucial for tasks such as reading and driving. Vision loss results from the death of optic nerve cells responsible for transmitting visual information to the brain. While central vision may gradually improve in a small percentage of cases, for most, the vision loss is severe and irreversible. LHON is triggered by mutations in the MT-ND1, MT-ND4, MT-ND4L, or MT-ND6 genes within mitochondrial DNA (mtDNA), which is distinct from nuclear DNA. These genes guide the production of proteins essential for mitochondrial function that are involved in converting oxygen, fats, and sugars into energy. Mutations disrupt this process, causing optic nerve cell death and LHON features [38]. The exact mechanisms behind how these genetic changes induce LHON symptoms remain unclear. Interestingly, a significant percentage of individuals with LHON-associated mutations may remain asymptomatic. More than 50 percent of males and over 85 percent of females with a mutation may not experience vision loss or related health issues. Environmental factors such as smoking and alcohol use are under investigation, with conflicting study results. Researchers are also exploring the role of additional gene changes in LHON symptom development.

Since oxidative stress is among the main culprits of LHON, antioxidant treatments have been used to fight disease progression [39]. Clinically, oral treatment with the antioxidant idebenone is the main approach to delaying disease progression [40]. Indeed, Raxone^®^ was approved by the FDA as an oral treatment for LHON [41].

**Stargardt macular dystrophy** (STGD) is an inherited eye disorder characterized by a gradual loss of vision [42]. This prevalent form of juvenile macular degeneration typically manifests its signs and symptoms from late childhood to early adulthood, progressing over time. The prevalence of STGD is estimated to be around 1 in 9,000 individuals. The condition specifically impacts the macula, where the fatty yellow pigment called lipofuscin accumulates in the retinal pigment epithelium (RPE) cells. Due to impaired recycling and disposal mechanisms, lipofuscin accumulates with toxic effects, damaging rod and cone photoreceptors in an area crucial for clear vision. Over time, this abnormal buildup results in issues such as impaired night vision, making it challenging to navigate in low light. Some individuals also experience difficulties with color vision. The primary cause of Stargardt macular degeneration is often attributed to variants or mutations in the ABCA4 gene, which is responsible for producing a protein involved in transporting potentially harmful substances out of light-sensing cells (photoreceptors) in the retina. Variants in the ABCA4 gene hinder the protein’s ability to remove toxic byproducts, resulting in the accumulation of lipofuscin and subsequent cell death in the retina, leading to progressive vision loss [43]. Photo-oxidative damage to RPE and photoreceptors may derive from the buildup of a substance known as A2E within RPE cells, formed through the condensation of phosphatidylethanolamine (PE) and the all-trans-retinal released from photoactivated rhodopsin [44]. A2E, acting as a photosensitizer, generates reactive singlet oxygen and undergoes photo-oxidation, leading to retinal damage and apoptosis in photoreceptor and RPE cells. The resulting photo-oxidation products of A2E activate the complement system, causing inflammation and DNA damage in RPE cells. A2E, along with its photo-oxidized products, including methylglyoxal, disrupts mitochondrial dynamics and function, ultimately inducing apoptosis in RPE cells. Photosensitization of A2E expedites the production of inflammatory cytokines, contributing to DNA damage, telomere deletion, and triggering RPE senescence. Furthermore, the photo-oxidation of bisretinoids such as A2E enhances lipid peroxidation and upregulates the expression of oxidative stress and complement activation genes, concurrently downregulating protective complement regulatory proteins [45]. Therefore, antioxidant treatment with xanthophylls such as lutein and zeaxanthin, which are naturally present in the macula, and with other exogenous antioxidants might also play a role in STGD, attenuating the symptoms and the progression of the pathology. In fact, a clinical study in which patients were orally treated with the antioxidant saffron showed a stabilization of the disease over the 6 months of treatment, while placebo-controlled patients showed further progression [46]. Moreover, the presence of melanin, also endowed with antioxidant potential, in RPE cells may contribute to the endogenous free radical scavenging activity of this tissue, contrasting the insurgence and progression of STGD [47].

To date, there are two possible ways to fight these hereditary retinal dystrophies. The first, more challenging but with great expectations, is gene therapy, aiming at the correction of the genetic mutations that cause the pathology [48]. The second, easier in its application, already practiced and under continuous evaluation, is based on the treatment with antioxidants [49].

Antioxidant treatments are usually given by the oral systemic route. For instance, the only approved antioxidant treatment for LHON (Raxone^®^) is taken by the oral route. Similarly, nutraceutical treatment of RP (NUTRARET: see above) occurs through the oral route. Given the fact that the eye remains peripheral to the systemic circulation and represents only a small part of the body, high doses are necessary in order to reach an effective concentration within the eye and at the retinal level. Moreover, the blood–retinal barrier may limit the types of molecules that can reach the retina. Topical treatment would be much better, bringing the treatment in close proximity to the retina. However, eye injections can be used for treatments given at an interval of months; otherwise, they become too risky for the health of the eye. The other option is topical treatment with eye drops. In this case, the major obstacle is the epithelial barrier of the cornea and conjunctiva, which limits the number of molecules able to cross it and progress all the way to the retina. Appropriate formulations are required in order to enhance delivery through the epithelial barriers [50,51].

## 3. Melatonin

Melatonin, a hormone produced by the pineal gland in the brain in response to darkness, plays a crucial role in regulating the sleep–wake cycle. Beyond its well-known function in sleep regulation, melatonin has been the subject of studies exploring its antioxidant and neuroprotective properties [52]. In terms of antioxidant activity, melatonin serves as a potent free radical scavenger, neutralizing harmful molecules that can damage cells and contribute to various diseases and aging. It also stimulates the activity of key antioxidant enzymes, such as superoxide dismutase (SOD), glutathione peroxidase, and catalase, which are vital for the body’s defense against oxidative damage. Moreover, melatonin possesses metal-chelating properties, binding to and neutralizing metal ions that contribute to oxidative stress, thereby preventing their involvement in reactions that generate free radicals [53]. Melatonin is also endowed with neuroprotective and anti-inflammatory activities. However, the role of melatonin in oxidative stress is controversial, as recently highlighted by Boutin and collaborators. In particular, these authors question the hypothesis that melatonin acts as a scavenger, reporting that much of the experimental evidence is sometimes imperfect. In fact, the evidence from clinical trials on the efficacy of melatonin is mixed. Some studies have shown that melatonin is effective for treating conditions like insomnia, jet lag, or AMD, while others have found no significant benefits. Nevertheless, the majority of studies have indicated that melatonin is safe and well tolerated [54]. Generally speaking, it is conceivable that the antioxidant properties of melatonin play a role in reducing oxidative stress, a common factor in the development and progression of neurodegenerative diseases. Additionally, melatonin exhibits anti-inflammatory effects, which are crucial given the implications of chronic inflammation in various neurodegenerative conditions. The hormone’s ability to modulate inflammatory pathways contributes to its neuroprotective effects [55,56]. Melatonin has also been shown to enhance neuroplasticity, the brain’s ability to reorganize itself by forming new neural connections, which is fundamentally significant for learning and memory processes [56]. Furthermore, melatonin reportedly protects mitochondria from impairment. In fact, by preventing mtDNA damage, retaining mitochondrial ATP synthesis, and re-establishing the mitochondrial respiratory control system, melatonin may preserve mitochondrial homeostasis.

Given that mitochondrial dysfunction is associated with various neurodegenerative disorders, melatonin’s protective effects on mitochondria contribute to its overall neuroprotective properties [57].

Beyond these primary roles, melatonin presents other potential benefits. It has been found to have anti-apoptotic effects, preventing cell death and potentially benefiting the preservation of neurons and other cells in the CNS, including retinal cells [58].

In addition, melatonin facilitates physiological vascular endothelial growth factor (VEGF) production in the retina, thus preventing neovascularization [59].

The multifaceted antioxidant and neuroprotective activities of melatonin highlight its potential as a therapeutic agent in addressing various conditions related to oxidative stress and neurodegeneration.

In fact, melatonin has been suggested for the treatment of some age-related ocular diseases, including AMD, DR, cataracts based on oxidative stress [60], and immunologic ocular diseases such as uveitis [61].

A formulation of melatonin as eye drops in a Tris-buffered nanomicellar preparation has been recently described. In this way, it can penetrate the eye and diffuse into the ocular surface, also reaching the anterior chamber, the vitreous, and the retina. It has been shown that after topical administration, melatonin eye drops could decrease IOP and attenuate the signs of glaucoma in both normal rats and rats in which ocular hypertension had been induced by artificial clogging of the trabecular meshwork [62,63]. Therefore, it can be supposed that using such a transepithelial formulation, the multifaceted properties of melatonin could be used to protect the different structures of the eye reached by this molecule, such as the cornea, the trabecular meshwork in the anterior chamber, the lens, and, finally, the retina [64]. Moreover, melatonin can protect the retina in mouse models of RP [65,66], while no data exist for an effect of melatonin in LHON or STGD. However, absence of proof is not proof of absence; therefore, it would be interesting to see whether melatonin eye drops may also exert some protective effect in these retinal dystrophies.

## 4. N-Acetyl-Cysteine

N-acetyl-cysteine (NAC), an acetylated compound derived from cysteine, is known for its antioxidant properties and has been extensively studied in different model systems [67]. Recently, there has been a growing interest in exploring the potential benefits of NAC in ocular diseases, driven by its ability to counteract oxidative stress [68]. Functioning as a precursor to glutathione, a potent endogenous antioxidant, NAC neutralizes ROS and free radicals implicated in oxidative stress related to various ocular diseases. Moreover, NAC is characterized by the presence of a nucleophilic free sulfhydryl group, which promotes direct antioxidant activity. In fact, NAC is able to counteract the inactivation of α1-antitrypsin by hypochlorous acid, which is produced by activated phagocytes [69]. Another role of NAC is to act as an anti-inflammatory factor, inhibiting cytokine release and nuclear factor kappa-B (NF-κB) expression, which regulates pro-inflammatory genes involved in the immune response [70]. Several studies on the topical use of NAC in the eye also show an improvement in epithelial wound healing. An in vitro study with human corneal epithelial cells shows that NAC inhibits the production of matrix metalloproteinase-9 (MMP9), which is upregulated in disorders of the ocular surface. Sekundo and colleagues demonstrated that NAC eye drops improve healing in corneal injuries, likely due to their potent collagenase and scavenger potential [71]. Therefore, NAC’s antioxidant and anti-inflammatory properties are considered in addressing dry eye syndrome, characterized by inflammation and oxidative stress on the ocular surface [70]. N-acetyl-cysteine amide (NACA) eye drops enhancing GSH levels prevented selenite-induced cataracts in a rat experimental model [72]. The potential of NAC in combination therapies with other antioxidants or therapeutic agents is also under scrutiny, aiming to enhance its efficacy in addressing oxidative stress in ocular diseases.

The lens, too, susceptible to oxidative damage leading to cataracts, may benefit from NAC by replenishing glutathione levels and supporting antioxidant defense mechanisms [73].

Furthermore, NAC’s neuroprotective effects in the retina after oral administration are under investigation for potential applications in conditions such as age-related macular degeneration (AMD) [74] and diabetic retinopathy [75], where oxidative damage is a contributing factor. In the context of glaucoma, NAC is being explored for its role in managing oxidative stress associated with increased intraocular pressure and optic nerve damage [76]. Finally, NAC might also contribute to contrasting the altered metabolic pathways at the root of RP [77], LHON [78], and STGD [79]. Indeed, a multicenter, placebo-controlled clinical trial to test whether oral NAC can slow the progression of RP (NAC Attack, enrolling 438 participants) is running at the Wilmer Eye Institute of the Johns Hopkins University School of Medicine in Baltimore (MD, USA) [80].

Cysteine derivatives with strong antioxidant properties have been tested for ophthalmological treatments. For example, S-Allyl-L-cysteine (SAC) protects the retina of murine models from various insults, such as increased intraocular pressure and kainate excitotoxicity [81]. ARPE-19 cells treated with another derivative, S-allylmercapto-N-acetyl-cysteine (ASSNAC), show an increase in GSH and are protected from damage by hydrogen peroxide in comparison to cells treated with NAC. Even pre-incubation with ASSNAC inhibits the hydrogen peroxide-induced opacification of the swine lens [82]. Therefore, the development of more efficient transepithelial eye drop formulations of NAC could potentially serve as a powerful tool in combating oxidative diseases of the eye.

## 5. The Challenge of Drug Delivery

Lipid-based nanoparticles have been extensively used in drug delivery to the posterior segment of the eye, offering a promising approach to overcome the anatomical and physiological barriers that challenge the effective delivery of therapeutic agents to this specific region. These nanoparticles, often in the form of liposomes, lipid nanocarriers, or solid lipid nanoparticles, have unique characteristics that make them suitable for ocular drug delivery [83]. One key advantage is their ability to encapsulate both hydrophilic and hydrophobic drugs, allowing for the delivery of a wide range of therapeutic compounds. Lipid-based nanoparticles can be tailored to improve drug solubility, stability, and bioavailability, ensuring the efficient transport of drugs to the posterior segment. Moreover, these nanoparticles enhance drug penetration and residence time in ocular tissues. Their small size allows for increased bioavailability and sustained release of drugs, reducing the need for frequent administrations. The lipid-based nature of these nanoparticles also facilitates interactions with cell membranes, promoting cellular uptake and improving drug delivery to target cells within the posterior segment. Lipid-based nanocarriers can be engineered to provide controlled drug release, responding to various stimuli such as pH, temperature, or enzymatic activity. This controlled release feature is particularly advantageous for managing chronic ocular conditions, ensuring a prolonged therapeutic effect with minimum side effects [84]. In summary, lipid-based nanocarriers may serve as effective drug delivery systems to the posterior segment of the eye by addressing challenges associated with drug solubility, bioavailability, and tissue penetration. Their versatility in encapsulating different types of drugs, coupled with the potential for controlled release, makes them a valuable tool in developing innovative therapeutic strategies for ocular diseases affecting the posterior segment.

Based on these premises, we have developed functionalized antioxidant molecules with the aim of facilitating their insertion into lipid nanoparticles to be formulated for topical drug delivery.

## 6. Idebenone and Edaravone

***Idebenone*** is a short-chain benzoquinone that bears a structural resemblance to coenzyme Q10. Idebenone’s antioxidant properties are unique. While it does have direct antioxidant effects, it also enhances cellular mechanisms to counteract ROS. Idebenone upregulates the expression of endogenous antioxidants, such as superoxide dismutase (SOD), catalase, and glutathione peroxidase, while downregulating NOX2 [85]. Additionally, it restores mitochondrial function by stimulating respiration and ATP synthesis, particularly in cells lacking complex I [86]. IDB prevents lipid peroxidation, protecting the lipid membrane and mitochondria from oxidative damage [87]. IDB is cytoprotective, affecting nuclear factor-erythroid 2-related factor 2 (Nrf-2), a crucial transcription factor for maintaining cellular redox homeostasis. Nrf-2 induces the expression of genes encoding antioxidant proteins and phase 2 enzymes needed for the detoxification of xenobiotics. In basal conditions, Nrf-2 is related to Keap-1, which is its main negative regulator. Following electrophilic or oxidative stress, Nrf-2 detaches from Keap-1 and translocates within the nucleus, forming a heterodimer, binding to ARE sequences present on the DNA, and activating the transcription of its target antioxidant genes [88]. It has been shown that IDB exerts its cytoprotective effect on RPE cells exposed to oxidative stress by reducing intra-cytoplasmic ROS levels and normalizing the Bax/Bcl-2 ratio [89].

Because of its properties, idebenone has garnered attention for its potential therapeutic applications in various human pathologies. Its primary focus lies in addressing conditions linked to mitochondrial dysfunction and oxidative stress.

Idebenone has been explored in the context of Friedreich’s ataxia, a genetic disorder marked by progressive nervous system damage. Studies have delved into its impact on cardiac function and its role in reducing oxidative stress, suggesting potential benefits in mitigating the effects of this disorder [90]. In the realm of neurodegenerative diseases, such as Alzheimer’s and Parkinson’s, idebenone’s neuroprotective properties have been investigated. Its antioxidant capabilities and influence on mitochondrial function make it a subject of interest in conditions marked by oxidative stress and mitochondrial dysfunction [91]. Duchenne muscular dystrophy (DMD), a genetic disorder characterized by progressive muscle degeneration, has also been focused on by idebenone research. Some studies suggest potential benefits in terms of respiratory function and overall quality of life for individuals with DMD [92].

Mitochondrial dysfunction is also implicated in various eye pathologies, making idebenone a promising candidate for therapeutic interventions. In fact, oxidative stress plays a crucial role in the pathogenesis of many eye diseases, including AMD, DR, and glaucoma. Research on patients with AMD has shown promising results, suggesting that idebenone supplementation may slow down the progression of the disease and improve visual function [93]. Additionally, idebenone has demonstrated potential benefits for improving the survival of retinal cells in experimental glaucoma models [94].

In clinical trials, idebenone has demonstrated efficacy in the recovery of lost vision and the preservation of good residual vision among patients with LHON. LHON is a genetic disorder affecting the optic nerve, and idebenone’s ability to mitigate the effects of mitochondrial dysfunction in this context has significant implications for the treatment of this condition [95]. While research is ongoing, more recent clinical findings suggest that idebenone supplementation may offer a potential avenue for slowing vision deterioration in individuals affected by LHON [96].

Currently, Raxone (idebenone) is the only authorized medicinal product within the European Union for the treatment of LHON. Available evidence suggests a potential benefit for vision in a specific group of individuals treated with idebenone, particularly those receiving treatment early in the disease course [40,95]. Therefore, despite this approval, LHON continues to be an unaddressed medical necessity, emphasizing the limited options available for effectively managing this condition within the EU.

In summary, idebenone’s unique mechanism of action, enhancing cellular antioxidant defenses and restoring mitochondrial function, positions it as a potential therapeutic agent for various eye pathologies, especially those associated with mitochondrial dysfunction. The positive outcomes observed in clinical trials, particularly in LHON patients, highlight its promising role in addressing vision-related disorders. More recently, topical formulations of idebenone have been proposed using either lipid nanoparticles [97] or β-cyclodextrins [98] as carriers.

***Edaravone***, recognized for its role as an oxygen-free radical scavenger, has demonstrated potential as an antioxidant treatment for retinal diseases. Its application extends to the treatment of acute ischemic stroke, where its ability to reduce oxidative stress and eliminate free radicals contributes to neuroprotection [99]. More studies have investigated its potential benefits in addressing cerebrovascular ischemia and cerebral edema, conditions linked to oxidative stress [100].

Beyond its direct antioxidant action, edaravone exhibits neuroprotective effects in the retina; glaucoma, DR, AMD, and retinal vein occlusion (RVO) have been shown to benefit from edaravone treatment [101]. Edaravone’s influence on inflammatory processes in ocular tissues is noteworthy, with its ability to alleviate inflammatory activation and modulate the expression of various inflammatory markers such as TNF-α, IL-1β, IL-6, and NOS2. In addition, it has been found that edaravone reduces hyperosmolarity-induced toxicity in culture models of human corneal epithelial cells (HCECs). It improved mitochondrial dysfunction, attenuated hyperosmolarity-induced cell death and apoptosis, and augmented the expression of Nrf2 and its target genes, such as hemeoxygenase 1 (HO-1), glutathione peroxidase (GPx), and glutamate cysteine ligase (GCLC) [102]. Consideration has also been given to the topical application of edaravone, aiming to target ocular tissues directly [103,104]. However, challenges related to drug delivery efficiency, bioavailability, and interactions with eye structures necessitate careful consideration.

## 7. Functionalization of Edaravone

Given the strong antioxidant potential and neuroprotective efficacy of edaravone, and since its effects on retinal diseases have not been explored yet, we used edaravone as a prototype molecule for functionalization studies in order to facilitate its insertion into lipid nanoparticles. Therefore, in an effort to enhance the antioxidant efficacy of edaravone within living systems, particularly in lipid-rich compartments, a C18 hydrocarbon chain was introduced at the C-4 position of its pyrazolone ring [105] (Figure 1).

The resulting compound, C18-EDV, displayed unaltered radical scavenging activity; however, it also exhibited a significant improvement in safeguarding against lipid peroxidation compared to EDV. To elaborate further, the inclusion of the lipophilic C18 chain enhanced protection within bilayers when radicals originated in the lipid phase using 2,2-azobis(2,4-dimethylvaleronitrile) (AMVN). Conversely, a reduction in protection was noted when radicals were generated in the aqueous phase via the water-soluble initiator 2,2’-Azobis(2-amidinopropane) dihydrochloride (AAPH). This observation is attributed to the more effective integration of C18-EDV into the lipid matrix of bilayers compared to EDV. The robust interaction between the hydrophobic carbon chain of lipidic tails in membranes and C18-EDV enhances its affinity for cellular membranes, thereby augmenting its capacity to safeguard retinal cells [95]. Further studies were carried out to establish the best conditions to allow the proper insertion of the lipophilic derivative of edaravone within the lipid bilayer of liposomes. From the combined results of two studies [106,107], it was found that the combination of C18-EDV at 20% *w*/*w* of antioxidant concentration and 0.25 M of CaCl_2_ was the best formulation to be used as a starting point to protect membranes from external free radicals. These findings imply that C18-EDV may function effectively as an antioxidant in both cellular and cell-free systems, showing promise as a prospective therapeutic drug candidate for conditions associated with oxidative stress. It will be interesting to study its PK and PD after topical administration as an eye drop formulation in animal models of LHON [108] or other different eye diseases in which oxidative stress plays a prominent role.

## 8. Epigallocatechin-3-Gallate

Epigallocatechin-3-gallate (EGCG), a polyphenol found in abundance in green tea, has emerged as a subject of intense scientific interest because of its potential beneficial effects on human health [109]. This compound, belonging to the family of catechins, possesses powerful antioxidant and anti-inflammatory properties that extend beyond its well-known role as a beverage ingredient. In recent years, research has highlighted the diverse ways in which EGCG may positively impact various aspects of human health, with a particular focus on its potential benefits for ocular well-being.

One of the key attributes of EGCG lies in its potent antioxidant activity [110]. As an efficient scavenger of free radicals, EGCG helps neutralize ROS and mitigate oxidative stress. Oxidative stress is implicated in the pathogenesis of numerous health conditions, including age-related eye diseases. The retina is vulnerable to oxidative damage, inflammation, and other stressors. EGCG’s anti-inflammatory properties contribute to modulating pathways involved in retinal inflammation, thus potentially offering protective effects against inflammatory-driven eye diseases. In the context of eye health, EGCG’s ability to counteract oxidative damage holds promise for conditions such as age-related macular degeneration (AMD), DR, and glaucoma, where inflammation and oxidative stress play a significant role in disease progression [111]. More studies have suggested that EGCG may exert protective effects on retinal cells and tissues [112,113]. Moreover, EGCG has been investigated for its neuroprotective effects on the optic nerve, which is critical for transmitting visual information from the eye to the brain, and its health is paramount for maintaining vision. Some research indicates that EGCG may contribute to the preservation of optic nerve function, making it a potential candidate for conditions involving optic nerve damage [114]. The potential benefits of EGCG extend beyond its antioxidant and anti-inflammatory properties. EGCG has demonstrated antiangiogenic effects, meaning it may help inhibit the formation of abnormal blood vessels in the eye. In conditions such as DR, abnormal blood vessel growth can lead to vision impairment. EGCG’s ability to modulate angiogenesis represents a promising avenue for therapeutic interventions in such cases [115,116]. Furthermore, EGCG’s impact on cellular signaling pathways and its ability to regulate gene expression have been subjects of interest in the context of eye health [117]. These molecular mechanisms may contribute to EGCG’s multifaceted effects, influencing cellular processes involved in maintaining ocular homeostasis.

EGCG is able to modulate autophagy in many cellular systems. Autophagy allows cells under oxidative stress to survive through its interaction with apoptosis and the MTOR-kinase AMP-activated alpha 1 catalytic subunit (PRKAA1/AMPK) [118]. It has been shown that EGCG promotes autophagy by forming autophagosomes and inducing lysosomal acidification and autophagic flow in Müller retinal cells. Retinal degeneration (RD) is a characteristic of DR, characterized by a high concentration of glucose in the retina [119]. An increase in glucose has been associated with an inhibition of autophagy due to the accumulation of sequestosome 1 (p62/SQSTM1) load, the downregulation of beclin 1 (BECN1), and increased apoptosis. In an experimental model of DR, EGCG reduces the damage to the retina induced by glucose. The observed effect depended on the activating kinase MTOR and unc-51-like autophagy (ULK1) [120].

Therefore, the beneficial effects of EGCG on human health, particularly in the realm of eye health, underscore its potential as a natural and accessible therapeutic agent. The antioxidant, anti-inflammatory, and neuroprotective properties of EGCG position it as a promising agent for future research and may pave the way for innovative strategies for preventing and managing various eye conditions.

Gelatin-EGCG nanoparticles coated with hyaluronic acid (HA) have been proposed to deliver EGCG to the retina after topical administration (as eye drops) or subconjunctival injection. Fluorescently labeled NPs were traced using whole-eyeball cryosections. The area of fluorescent signal in the posterior eyes treated with GEH + NPs was notably higher in both delivery methods (eye drops: 6.89% and SCI: 14.55%) compared to other groups, particularly surpassing the free dye solution group (2.79%) [121].

## 9. Functionalization of EGCG

Similar to the approach taken with EDV, EGCG was also alkylated with a C18 chain to facilitate its insertion into the double lipid layer of liposomes [122] (Figure 2).

The alkylation consisted of the introduction of an aliphatic C18 chain linked to the gallate ring (circled in Figure 2) through an ethereal bond, as determined by NMR and confirmed by Density Functional Theory (DFT) calculations. The antioxidant activity of the mono-alkylated EGCG (C18-EGCG) was evaluated through DPPH and TBARS assays, and its protective effects against oxidative stress were assessed in ARPE-19 cells. Molecular Dynamics (MD) simulation and liposomal/buffer partition studies were employed to investigate the interaction of the modified and unmodified antioxidants with a cell membrane model. The combined experimental and in silico approaches revealed the increased affinity of C18-EGCG toward the lipid bilayer. While the DPPH assay indicated a decrease in EGCG activity against free radicals due to functionalization, cellular experiments demonstrated that the lipid moiety heightened the antioxidant protection of the lipophilic derivative. The introduction of a C18 chain in the EGCG structure primarily resulted in EGCG monoalkylated in the 4” position of ring D. Despite the fact that the EGCG derivative showed lower DPPH radical scavenging activity compared to EGCG, C18-EGCG exhibited an enhanced ability to protect retinal cells. This outcome can be attributed to the unique structure of C18-EGCG, where increased lipophilicity enhances its affinity for the cellular membrane. The presence of the gallate moiety in close proximity to phosphate groups allows the antioxidant to intercept radicals, particularly targeting the aliphatic chain within the lipid system [122].

It will be intriguing to study whether a liposomal eye drop formulation containing C18-EGCG would be able to permeate the eye and treat animal models of pathological conditions associated with oxidative stress and inflammation, such as those described in this paper.

## 10. Discussion

So far, most inherited retinal diseases lack a cure. Some hope may lie in AAV-gene therapy, which recently resulted in the approval of voretigene neparvovec (Luxturna^®^) for patients with biallelic RPE65 mutation-associated inherited retinal dystrophy [123]. However, the eyes, due to their unique physiological structure and constant exposure to the external environment, are susceptible to damage from ROS, which tend to accumulate in the retina and lens structures. The generation of ROS can be attributed to both endogenous factors, particularly the mitochondrial respiratory chain, and exogenous factors such as environmental pollution, radiation, UV exposure, certain foods, and medications. The presence of oxidative stress is closely linked to the onset of various diseases.

Crucially, antioxidant therapies play a significant role in mitigating oxidative stress by indirectly addressing the underlying mechanisms that lead to imbalance (Table 1). Antioxidant therapies, either through supplementation or pharmacological inhibitors, have demonstrated potential in the early stages of various diseases, as evidenced in recent studies [124,125]. In the realm of ocular therapeutics, the emphasis is on enhancing the bioavailability of topical treatments. The carriers used in these therapies, often formed through the spontaneous assembly of their components in water, require minimal steps and energy for their creation. The utilization of nanocarriers in the formulation and topical delivery of therapeutics via eye drops represents a promising avenue in ocular drug delivery. Nanocarriers, such as nanoparticles or liposomes, provide a versatile platform for enhancing the solubility, stability, and bioavailability of drugs [126]. Their small size facilitates efficient penetration into ocular tissues, allowing for targeted drug delivery to specific cells or structures within the eye. The use of nanocarriers in eye drop formulations minimizes the need for frequent administrations and reduces systemic side effects. Additionally, the controlled release capabilities of nanocarriers contribute to prolonged therapeutic effects. This approach has shown potential for treating various ocular conditions, including glaucoma, macular degeneration, diabetic retinopathy, and other inflammatory eye diseases [127]. As research in nanotechnology continues to advance, the development of innovative nanocarrier-based formulations holds great promise for improving the efficacy and precision of ocular drug delivery [128].

Moreover, as our understanding of the molecular mechanisms underlying various retinopathies increases and the pivotal role played by mitochondria in most oxidative stress-derived ophthalmic pathologies becomes evident, new targets come to the attention of scientists working on the development of novel and more effective therapeutic drugs [129]. These drugs could be directly developed and formulated for delivery through topical nanotechnological methods [130].

## 11. Conclusions

In line with the expected developments in drug formulation and delivery, the findings reported in our review suggest that introducing a carbon chain, specifically C18, may open new avenues for treating ocular diseases. This addition not only enhances the passage of therapeutic agents but might also improve their therapeutic effects. The use of such modified structures can potentially revolutionize ocular drug delivery and improve the overall efficacy of treatments for various eye conditions. Of course, these data have to be considered preliminary with respect to possible therapeutic applications in humans because they still have to pass further steps in development, showing efficacy in experimental model systems in vivo and, finally, in clinical trials. Nonetheless, they appear to be a promising new approach to enhance the activity of old drugs or to improve the delivery of newly developed therapeutic molecules.

## Figures and Tables

**Figure 1 medicina-60-00354-f001:**
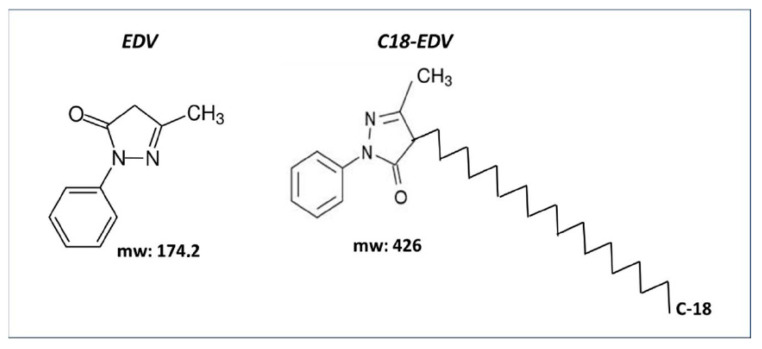
Edaravone structure and C18-functionalized edaravone.

**Figure 2 medicina-60-00354-f002:**
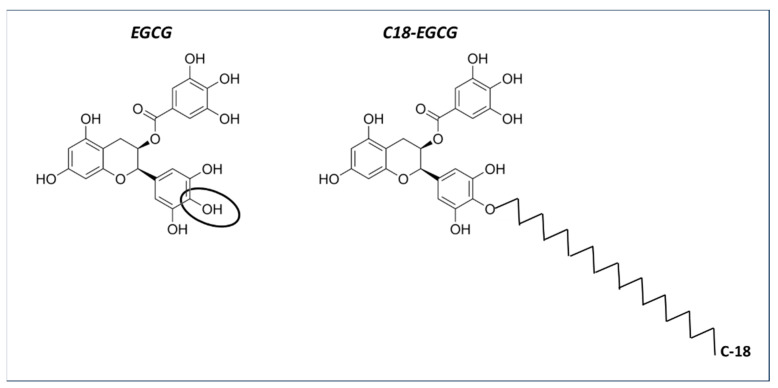
The structure of epigallocatechin-3-gallate and C18-functionalized epigallocatechin-3-gallate.

**Table 1 medicina-60-00354-t001:** Activity and effects of antioxidant compounds in different ophthalmic diseases based on experimental studies.

Antioxidant Compounds	Properties	Effect on Eye Structures	Potential Utilization in Ophthalmic Pathologies	Type of Study and Results	References
Melatonin 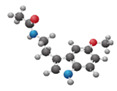	GPX, SOD, and catalase induction Anti-inflammatory Mitochondrial homeostasis Metal-chelating Anti-apoptotic Neuroprotective Enhances neuroplasticity Prevents neovascularization	Retina Cornea Trabecular meshwork Anterior chamber Lens	IOP Glaucoma AMD DR Cataract	In vivo study (rats). Melatonin showed potential neuroprotective properties by increasing the release of neurotransmitters, antioxidants, and anti-inflammatory factors and reducing pro-inflammatory cytokines and apoptosis in the brain. In vitro study (RPE). Melatonin decreased H_2_O_2_-induced apoptosis in RPE cells. In vivo study (C57BL/6J mice). Melatonin restored visual functions and reversed the decrease in RPE melanin content and RPE65 immunoreactivity. In vivo study (hamsters). Melatonin treatment started after the onset of uveitis attenuated ocular inflammation induced by LPS. In vivo study (rats). Melatonin formulated in nanomicelles had a longer-lasting hypotonizing effect on IOP. In vivo study (mouse). Melatonin delayed photoreceptor degeneration in rds/rds mice.	[56,59,60,61,62,63]
NAC 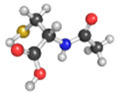	Neutralizes. α1-antitrypsin inactivation Anti-inflammatory Inhibits MMP9 Neuroprotective	Retina Corneal epithelial cells Ocular surface Lens	Epithelial wound-healing Corneal injuries Dry eye syndrome AMD DR RP IOP Glaucoma Cataract LHON STGD	In vivo study (rats). NAC eye drops administration in Wistar rat pups indicated that NAC was able to reverse the cataract grade. In vivo study (rats, pilot study). NAC produced a protective mechanism against IVTA-induced cataract. In vivo study (rats). NAC treatment alleviated the pathological changes and decreased ROS in diabetic rats. In vivo study (mouse): NAC treatment in excitatory amino-acid carrier 1 (EAAC1) KO mice. Oxidative stress and autophagy were suppressed and glutathione levels increased. In vitro (ARPE-19 human RPE cells). S-Allylmercapto-N-acetyl-cysteine (ASSNAC) upregulated glutathione levels, protecting the cells from oxidative stress-induced cell death and protecting lenses from oxidative stress-induced opacity.	[72,73,75,76,82]
Idebenone 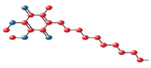	GPX, SOD, and catalase induction Prevents lipid peroxidation Downregulates NOX2 Mitochondrial homeostasis Neuroprotective Anti-apoptotic	Retina	AMD DR Glaucoma LHON	(Clinical Trial). LHON patients treated with idebenone showed vision improvement. In vitro study (HQB17, RJ206, and XTC.UC1 cells). Idebenone activity on mitochondrial function. In vitro (ARPE-19 human RPE cells). Idebenone increased ARPE-19 cell survival and reduced cell death, senescence, and oxidative stress by stabilizing the BAX/Bcl-2 ratio. In vitro study (optic nerve head astrocytes). Idebenone reduced senescence, oxidative stress, and apoptotic cell death in cultured optic nerve head astrocytes. (Clinical Trial). Patients with discordant visual acuities in Leber’s hereditary optic neuropathy are the most likely to benefit from idebenone treatment, which is safe and well tolerated.	[40,86,93,95,97]
Edaravone 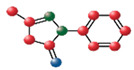	Anti-inflammatory Neuroprotective Mitochondrial homeostasis Reduces hyperosmolaity Anti-apoptotic	Retina Corneal epithelial cells	Glaucoma DR AMD Retinal Vein Occlusion Animal models of LHON	In vivo study (mice). Edaravone-loaded submicron-sized liposomes (ssLips) protected against light-induced retinal dysfunction by eye drop administration. In vitro study (ganglion cell layer cell). Edaravone-loaded ssLip based on egg phosphatidylcholine (EPC-ssLip) significantly reduced NMDA-induced ganglion cell layer cell death compared with free edaravone. In vitro study (adult RPE cells). C18-edaravone is able to contrast 2,2-azobis (2-amidinopropane hydrochloride) (AAPH)-induced cell death.	[103,104,105]
EGCG 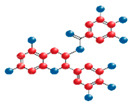	Anti-inflammatory Anti-angiogenesis Modulates autophagy	Retina Optic nerve	AMD DR Glaucoma	In vitro study (RGC-5 cells). EGCG protects retina neurons in situ from ischemia/reperfusion and from an oxidative stress insult. In vivo study (BALB/cJ mice). Optical coherence tomography (OCT), histology, electroretinography, and qPCR were used to examine how EGCG protected the retina of BALB/cJ mice by attenuating the detrimental effects of bright light. In vitro study (RGCs). Immunohistochemical and Western blotting analyses suggested the protective effects of EGCG on RGCs after optic nerve crush, indicating EGCG might be a potential treatment agent for optic nerve diseases.	[112,113,114]

## Data Availability

The data presented in this study are openly available.

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
