# Peer review of "Options for Topical Treatment of Oxidative Eye Diseases with a Special Focus on Retinopathies"

_medicina, 2024, doi:10.3390/medicina60030354_

Round 1
Reviewer 1 Report
Comments and Suggestions for Authors
The authors present an interesting review about the role of oxidants in the aetiopathogenesis of neurological and neuro-ophthalmological diseases, and the hypothesis that the use of antioxidants could have a potential benefit in eye diseases such as certain optic neuropathies, in which some studies have been carried out, and their possible therapeutic role in eye diseases such as glaucoma or age-related macular degeneration that are highly prevalent.
Several studies that have been cited are based on in vitro experimental models and animal experimental models.
There is not enough evidence to consider that these agents could have an application in clinical practice or may do so in the future. The development of nanoparticles that allow the intraocular release of these agents will make it possible to assess their usefulness in the ocular diseases mentioned in this paper. I believe that this should be specified in the discussion and in particular in the conclusion of this interesting review.
Author Response
Dear reviewer,
First of all, we would like to thank for reviewing the paper and suggesting further improvements.
Following your suggestions, please note that all the changes made to the manuscript have been highlighted in red.
Q.1
The authors present an interesting review about the role of oxidants in the etiopathogenesis of neurological and neuro-ophthalmological diseases, and the hypothesis that the use of antioxidants could have a potential benefit in eye diseases such as certain optic neuropathies, in which some studies have been carried out, and their possible therapeutic role in eye diseases such as glaucoma or age-related macular degeneration that are highly prevalent.
Several studies that have been cited are based on in vitro experimental models and animal experimental models.
There is not enough evidence to consider that these agents could have an application in clinical practice or may do so in the future. The development of nanoparticles that allow the intraocular release of these agents will make it possible to assess their usefulness in the ocular diseases mentioned in this paper. I believe that this should be specified in the discussion and in particular in the conclusion of this interesting review.
R.1
We thank the reviewer for the appreciation and the comments. We have explained better in the manuscript that all the data here mentioned have to be considered as preliminary, just suggesting possible avenues to improve treatment of the oxidative side of retinopathies (lines 104-109; 677-682).
We have added in the discussion several mentions of studies concerning the use of nanocarriers for the treatment of such pathologies (lines 641-659).

Reviewer 2 Report
Comments and Suggestions for Authors
I am just a simple ophthalmologist, not a biochemical scientist, but I have seen many patients with retinitis pigmentosa and other hereditary conditions desperately seeking for treatment. This article seems to give hope, but it is not justified hope. I have seen: high doses vitamin A, the “Cuba therapy”, ozone, blue berries, transcorneal electric stimulation, Dunaliella bardawilli (Israel), etcetera. After all these years, we only have a very expensive gene therapy for a very small group of patients (almost 400.000 euro’s per injection). The authors nevertheless pour all the possible medications over our heads without any critical word, they presume local therapy as eye drops might be more effective without any evidence. A great multicentre study for NAC in retinitis pigmentosa patients has been started by the John Hopkins University (including 438 patients), but is not mentioned. The study of Olivares-Gonzales et al. (2021) of only 31 patients, of whom 10 were treated for 2 years with a mixture of copper, zinc, selenium, lutein, zeaxanthin, vitamin A, B6 and B9, is cited as an example of “a promising avenue …. to preserve vision in individuals with RP “. Boutin et al. (2023; not cited) stress the lack of evidence that melatonin can be established as an in vivo radical scavenger! I am extremely interested and impatient to see an effective therapy for my patients, but this article is not pointing us the right way…
Comments on the Quality of English Languageno special comments
Author Response
Comments and Suggestions for Authors
Dear reviewer,
First of all, we would like to thank for reviewing the paper and suggesting further improvements.
Following your suggestions, please note that all the changes made to the manuscript have been highlighted in red.
Q.1
I am just a simple ophthalmologist, not a biochemical scientist, but I have seen many patients with retinitis pigmentosa and other hereditary conditions desperately seeking for treatment. This article seems to give hope, but it is not justified hope. I have seen: high doses vitamin A, the “Cuba therapy”, ozone, blue berries, transcorneal electric stimulation, Dunaliella bardawilli (Israel), etcetera. After all these years, we only have a very expensive gene therapy for a very small group of patients (almost 400.000 euro’s per injection). The authors nevertheless pour all the possible medications over our heads without any critical word, they presume local therapy as eye drops might be more effective without any evidence. A great multicentre study for NAC in retinitis pigmentosa patients has been started by the John Hopkins University (including 438 patients), but is not mentioned. The study of Olivares-Gonzales et al. (2021) of only 31 patients, of whom 10 were treated for 2 years with a mixture of copper, zinc, selenium, lutein, zeaxanthin, vitamin A, B6 and B9, is cited as an example of “a promising avenue …. to preserve vision in individuals with RP “. Boutin et al. (2023; not cited) stress the lack of evidence that melatonin can be established as an in vivo radical scavenger! I am extremely interested and impatient to see an effective therapy for my patients, but this article is not pointing us the right way.
R.1
We thank the reviewer for the useful comments, and for giving us his/her feelings as a field ophthalmologist. It is extremely important for researchers not to lose contact with those physicians who daily have to see and treat real patients, and decide what to do to improve their chances to heal, or at least not to progress in their disease. Also, we believe that prevention is a very important approach to patients at risk, and so the ophthalmologist should be well aware of treatments aimed at prevention or cure of the disease.
Specifically, to address the criticisms raised by this referee, we have made clear that all the data mentioned in this manuscript have to be considered as preliminary, just suggesting possible avenues to improve treatment or prevention of the oxidative side of retinopathies (lines 104-109; 677-682). We have mentioned where appropriate the clinical trial running at the Johns Hopkins Institute (lines 394-397), and the reference by Boutin et al. (lines 315-324). Gene therapy is mentioned in lines 625-627.

Reviewer 3 Report
Comments and Suggestions for Authors
The authors present a review of literature regarding the role of the oxidative stress in the eye diseases and the current therapeutic options. The subject is very interesting and the authors gather many novel informations on the topic.
However, the article should be better structured before publication:
1. More details regarding the role of the oxidative stress in AMD and DR should be added, while the presentation of genetic diseases could be shortened and focus on the topic.
2. The table with the anti-oxidative agents should be presented either in the beginning of discussion regarding the available compounds, or in the end. There is no sens to add it after the first compound is discussed.
The table should contain more data about the type of study( experimental, on cells/ animals, or clinical) and the brief results.
2. While therapies less used in clinical practice are discussed, the authors should not miss the most used compounds: lutein and zeaxanthin, omega-3 derived DHA and EPA, beta-carotene, vitamins A, C and E, and the antioxidative key role of microelements ( copper and Zinc).
3. A discussion paragraph, regarding the current limitations and future therapeutic targets.
Author Response
Comments and Suggestions for Authors
Dear reviewer,
First of all, we would like to thank for reviewing the paper and suggesting further improvements.
Following your suggestions, please note that all the changes made to the manuscript have been highlighted in red.
Q.1
The authors present a review of literature regarding the role of the oxidative stress in the eye diseases and the current therapeutic options. The subject is very interesting and the authors gather many novel information on the topic. However, the article should be better structured before publication:
Q.1 More details regarding the role of the oxidative stress in AMD and DR should be added
R.1 Done, see lines 129-149
Q.2 ..while the presentation of genetic diseases could be shortened and focus on the topic
R.2 Done, we have deleted some lines, as evidenced in the revised manuscript: lines 173-187; 220-225; 255-260.
Q.3 The table with the anti-oxidative agents should be presented either in the beginning of discussion regarding the available compounds, or in the end. There is no sense to add it after the first compound is discussed. The table should contain more data about the type of study( experimental, on cells/ animals, or clinical) and the brief results.
R.3 Done, the modified table has been moved to the end of the manuscript.
Q.4 While therapies less used in clinical practice are discussed, the authors should not miss the most used compounds: lutein and zeaxanthin, omega-3 derived DHA and EPA, beta-carotene, vitamins A, C and E, and the antioxidative key role of microelements ( copper and Zinc).
R.4 We have added new paragraphs dealing with all these compounds: lines 64-94
Q.5 A discussion paragraph, regarding the current limitations and future therapeutic targets.
R.5 Done, see lines 104-109; 677-682; 654-659

Round 2
Reviewer 2 Report
Comments and Suggestions for Authors
I find the article improved, but I still have a problem with the jubilant tone, which requires a lot of searching for answers by the reader. For instance, the authors state about LHON: "Clinically, oral treatment with the anti-oxidant idebenone is the main approach to delay disease progression [40]. Indeed, Raxone® was approved by the FDA as an oral treatment for LHON [41]." So, you would think that Raxone is available as treatment for LHON-patients, but only! patients with visual loss within the last year are allowed the therapy in specialized centres. About the results, Genereviews (2021) states: The current body of evidence indicates some visual benefit in a subgroup of affected individuals treated with idebenone, in particular those treated within the first year of disease onset [Klopstock et al 2011, Carelli et al 2011, Catarino et al 2020]. "Some visual benefit in a subgroup" sounds much more guarded... Also, the costs for one year of Raxone are more than € 7000, so it does not seem likely to be used for glaucoma, as the authors propose.
Comments on the Quality of English Language
I am not a native English speaker, but I only noticed just a minor imperfection.
Author Response
Dear reviewer,
First of all, we would like to thank for reviewing the paper and suggesting further improvements.
Following your suggestions, please note that all the changes made to the manuscript have been highlighted in red.
Q.
I find the article improved, but I still have a problem with the jubilant tone, which requires a lot of searching for answers by the reader. For instance, the authors state about LHON: "Clinically, oral treatment with the anti-oxidant idebenone is the main approach to delay disease progression [40]. Indeed, Raxone® was approved by the FDA as an oral treatment for LHON [41]." So, you would think that Raxone is available as treatment for LHON-patients, but only! patients with visual loss within the last year are allowed the therapy in specialized centres. About the results, Gene reviews (2021) states: The current body of evidence indicates some visual benefit in a subgroup of affected individuals treated with idebenone, in particular those treated within the first year of disease onset [Klopstock et al 2011, Carelli et al 2011, Catarino et al 2020]. "Some visual benefit in a subgroup" sounds much more guarded... Also, the costs for one year of Raxone are more than € 7000, so it does not seem likely to be used for glaucoma, as the authors propose.
R.
We thank the reviewer for the useful comments. We acknowledge the concerns of the referee, who, as a field ophthalmologist, naturally seeks practical approaches to improve patient care. However, this manuscript does not aim to provide immediate treatment recipes for retinopathies. Instead, it delves deeper into these pathologies, exploring their shared basis in oxidative stress and, consequently, proposing novel methods for topical antioxidant administration. The mention of idebenone (Raxone ®) serves solely to illustrate that systemic administration of an antioxidant represents the only currently approved therapy for LHON. The practical complexities associated with this treatment, such as limited efficacy, high cost, and restricted access, fall within the realms of social, political, and financial considerations. In order to comply with this indication by the referee, we have added the sentence in lines 481-483 (highlighted in blue).
The suggestion of using idebenone, or - more likely - other antioxidants in the treatment of glaucoma, is based on experimental evidence and the recognition of oxidative stress as a factor in glaucoma. It is not intended as an indication for field ophthalmologists to prescribe Raxone to glaucoma patients. Time, experience, and market forces will determine whether this option proves to be viable.

Reviewer 3 Report
Comments and Suggestions for Authors
The authors have revised the manuscript. I recommend accept.
Author Response
We thank the reviewer for the useful comment